# Species-specific identification of donkey-hide gelatin and its adulterants using marker peptides

Jinju Zhang[1,2], Menghua Wu[1,2], Zhiguo Ma[1,2], Ying Zhang[1,2]*, Hui Cao[1,2]*

1 College of Pharmacy, Research Center for Traditional Chinese Medicine of Lingnan (Southern China), National Engineering Research Center for Modernization of Traditional Chinese Medicine Lingnan Resources Branch, Jinan University, Guangzhou, Guangdong, China, 2 Guangdong Provincial Key Laboratory of Traditional Chinese Medicine Informatization (2021B1212040007), Guangzhou, Guangdong, China

* carolynzy@163.com (YZ); kovhuicao@aliyun.com (HC)

**Data Availability Statement:** All relevant data are within the article and its Supporting Information files.

**Funding:** This work was supported by Guangdong Natural Science Foundation (No. 2020A151501577, Ying Zhang), the Open Fund of

## Abstract

Donkey-hide gelatin is an important traditional Chinese medicine made from donkey skin. Despite decades of effort, identifying the animal materials (donkeys, horses, cattle and pigs) in donkey-hide gelatin remains challenging. In our study, we aimed to identify marker peptides of donkey-hide gelatin and its adulterants and develop a liquid chromatography–tandem mass spectrometry method to identify them. Theoretical marker peptides of four animals (donkeys, horses, cattle and pigs) were predicted and verified by proteomic experiments, and 12 species-specific marker peptides from donkey-hide gelatin and its adulterants were identified. One marker peptide for each gelatin was selected to develop the liquid chromatography–tandem mass spectrometry method. The applicability of the method was evaluated by investigating homemade mixed gelatin samples and commercial donkey-hide gelatin products. Using the liquid chromatography–tandem mass spectrometry method, the addition of cattle-hide gelatin and pig-hide gelatin to donkey-hide gelatin could be detected at a level of 0.1%. Horse-hide gelatin was detected when added at a level of 0.5%. Among 18 batches of donkey-hide gelatin products, nine were identified as authentic, and eight of the remaining samples were suspected to be adulterated with horse materials. These results provide both a practical method to control the quality of donkey-hide gelatin and a good reference for quality evaluations of other medicinal materials and foods containing protein components.

## Introduction

Donkey-hide gelatin (DHG), also known as Colla Corri Asini, has been used as a valuable medicine in China for thousands of years due to its effects such as nourishing the blood and enhancing physical abilities. DHG is a mixture of amino acids and high-molecular-weight polypeptides that are produced by the hydrolysis of collagens from donkey skin. However, the supply of donkey skins cannot fulfill the increasing demands for DHG due to donkey husbandry limitations. Therefore, skins from other animals, such as cattle, pigs and horses, are

Guangdong Province Key Laboratory of Pharmacodynamic Constituents of TCM and New Drugs Research (No. 2016B030301004, Hui Cao), and Cao Hui's Expert Workstation in Yunnan Province (202005AF150011, Hui Cao). The funder of Ying Zhang and Hui Cao had contributed to the project administration, supervision, validation, draft writing and editing throughout the period of this research.

**Competing interests:** The authors have declared that no competing interests exist.

often used illicitly as a substitute for DHG [1]. The safety and efficiency of DHG have been tested and verified by proven historical facts, but gelatins made from other animal skins may have unpredictable and serious consequences [2–4]. For example, it has been indicated that cattle-hide gelatins (CHGs) may cause safety problems because of the risk of bovine spongiform encephalopathy and foot-and-mouth disease [5, 6]. Pig-hide gelatins (PHGs) may present religious conflicts since the use of porcine foods and drugs is forbidden for people in some Islamic countries [7]. Furthermore, traditional Chinese medicine experts believe that DHGs and horse-hide gelatins (HHGs) have opposite effects.

Since DHG was listed in the Chinese Pharmacopoeia in 1977, great efforts have been dedicated to its identification. However, it is difficult to distinguish different gelatins, especially adulterated gelatins, using traditional physicochemical analysis methods [8–10]. Some widely used analytical methods, including polymerase chain reaction (PCR) and enzyme-linked immunosorbent assays (ELISAs), have been applied to identify gelatins [11, 12]. However, the thermal processing of gelatins at high temperatures may result in poor yield or degradation of DNA. This may lead to false and unreliable results and therefore, prevents PCR from being widely implemented. ELISA methods are also limited because of the protein degradation that can be caused by severe processing treatments [13]. Therefore, establishing an effective method to identify DHG and its adulterants is urgently needed.

Previous studies have proven that collagens, especially type I collagen, are the main active components in gelatins [14, 15]. Type I collagens contain two α1 chains [α1 (I)] and one α2 chain [α2 (I)]. Theoretically, the amino acid sequences of type I collagens vary from species to species. Therefore, we hypothesized that collagens from different animals may produce marker peptides that are unique to each species after being digested by trypsin [16] that may serve as markers to identify their corresponding gelatin. However, finding marker peptides in complex samples is very challenging. Fortunately, in recent years, highly sensitive and accurate liquid chromatography–tandem mass spectrometry (LC–MS/MS) proteomic profiling has led to the fast and accurate identification of peptides [16]. To date, several methods have been reported for identifying gelatins based on clustering analysis of MS/MS data [14, 17]. Other studies have predicted potential marker peptides by analyzing part of the type I collagen sequences followed by validation of these sequences using a proteomic method [10, 17, 18]. The latter strategy is better for the identification of adulterated gelatins and deserves further research.

In our study, both α1 (I) and α2 (I) were used to predict theoretical marker peptides. The presence of the theoretical peptides in gelatins was tested by LC–MS/MS proteomic analysis. Then, based on the detected marker peptides, an LC–MS/MS method was developed. The ability to identify DHG and its adulterants using this method was also evaluated. The study presented in this thesis is a comprehensive investigation to focus specifically on the identification of gelatins using marker peptides. Moreover, the method established here plays an important role in gelatin quality control.

## Materials and methods

### Samples and reagents

Dithiothreitol (DTT), 3-indoleacrylic acid (IAA), formic acid (FA), and acetonitrile (ACN) were purchased from Sigma (St. Louis, MO, USA). Water was purified using a Millipore purification system (Billerica, MA, USA). Sequencing-grade trypsin was obtained from Aladdin (Shanghai, China). A BCA protein assay kit was purchased from Beyotime (Nanjing, China). The four selected marker peptides were synthesized by Chinese Peptides (purity ≥ 95%, Shanghai, China).

Standard medicine samples of DHG (121274–201703), CHG (121695–201802) and PHG (121745–201701) were purchased from the National Institutes for Food and Drug Control (NIFDC; Beijing, China). Twenty-five batches of the four different gelatins (10 DHGs, 5 CHGs, 5 PHGs and 5 HHGs) were provided by the Inner Mongolia Gen-incept Biological Medicine Technology Co. (Inner Mongolia, China). All samples from the Inner Mongolia Gen-incept Biological Medicine Technology Co. were prepared according to the processing specifications in the Chinese Pharmacopoeia 2020 edition, and the quality of every gelatin sample complied with Chinese Pharmacopeia 2020 edition requirements [19]. A total of 18 batches of commercial DHG products were purchased from several pharmacies and medicine markets. The information for all of the samples collected in our study is detailed in S1 Table. No specific permissions were required for the collected samples as none of the studies involved animal research or sacrifice.

## Theoretical marker peptide prediction

To perform sequence alignment and predict theoretical marker peptides, collagen sequences, including α1 (I) and α2 (I) from donkeys, horses, cattle and pigs (accession numbers: B9VR88, B9VR89, F6SSG3, F6RTI8, P02453, P02465, A0A1S7J210, F1SFA7), were obtained from the Universal Protein Resource (UniProt, https://www.uniprot.org/). The collagen sequences from these four animal species were compared by sequence alignment using BioEdit software (version 7.0.9).

## Sample preparation for proteomic analysis

Three samples (one DHG, one CHG and one PHG) collected from the NIFDC and seven samples (two DHGs, two CHGs, two PHGs and one HHG) from the Inner Mongolia Gen-incept Biological Medicine Technology Co. were used for proteomic analysis. Sample preparation was performed via modified methods from previous studies [20, 21]. Each sample was ground in a crucible with a continuous supply of liquid nitrogen to create a powder. Then, 2 ml of RIPA lysis buffer was added to 0.5 g of accurately weighed powder to extract the protein. The product was concentrated using a 10 kDa centrifugal filter. Then, a BCA protein assay kit was used to determine the protein concentration in the supernatant according to the manufacturer's instructions. The protein sample (100 μg) was reduced using 10 mM DTT at 56˚C for 3 h. Next, alkylation was performed in 50 mM IAA for 40 min at room temperature in the dark. Then, the protein was digested with 3 μg of trypsin at 37˚C for 12 h. All digests were desalted using a SOLA HRP Cartridge (Thermo Fisher Scientific, USA) and dried with a vacuum freeze-dryer. The lyophilized peptides were reconstituted in 0.1% FA prepared for proteomic analysis.

## Proteomic analysis

To verify the predicted theoretical marker peptides, we performed proteomic experiments using an Orbitrap Fusion Lumos mass spectrometer (Thermo Fisher Scientific, USA) equipped with an EASY-nLC 1200 HPLC system (Thermo Fisher Scientific, USA). The HPLC and mass spectrometry (MS) instruments were both controlled by Xcalibur software (Thermo Fisher Scientific, USA). For LC separation, tryptic peptides were sequentially injected onto an Acclaim PepMap 100 C18 column (100 $\mu$m × 20 mm, 5 $\mu$m, Thermo, P/N: 1645664) and an Acclaim PepMap 100 C18 column (50 μm × 150 mm, 2 $\mu$m, Thermo, P/N: 164943). The mobile phases were Buffer A (2% ACN with 0.1% FA) and Buffer B (80% ACN with 0.1% FA). The following gradient program was utilized at a flow rate of 600 nL/min: 0–14 min, 5–12% Buffer B; 14–40 min, 12–24% Buffer B; 40–53 min, 24–38% Buffer B; 53–54 min, 38–95%

Buffer B; and 54−60 min, 95% Buffer B. The MS global settings were as follows: ion source type, nanospray ionization; spray voltage, 2.2 kV (positive); and capillary temperature, 270˚C. The MS parameters were as follows: detector type, orbitrap; orbitrap resolution, 60000 at 400 m/z; and mass precursor m/z range, 100.0−1500.0. The MS/MS parameters were as follows: product ion scan range, starting from m/z 100; activation type, CID; min. signal needed, 1500.0; isolation width, 3.00; normalized coll. energy; 40.0; default charge state, 6; activation Q, 0.250; activation time: 30.000; MS precursor m/z range, 50.0−1500.0; and data-dependent MS/MS, up to the 15 most intense peptide ions from the preview scan in the orbitrap.

## Database searching

The original MS/MS data were imported into Proteome Discoverer software (version 2.2) and searched against the UniProt *Equus caballus* database, UniProt *Equus asinus* database, UniProt *Bos taurus* database and UniProt *Sus scrofa* database. The search parameters were set as follows: mass analyzer, Orbitrap mass spectrometer; MS order, MS2; polarity mode, positive; enzyme name, trypsin (full); minimum peptide length, 6; precursor mass tolerance, 10 ppm; fragment mass tolerance, 0.02 Da; static modification, carbamidomethyl/+57.021 Da (C); and dynamic modifications (peptide terminus), oxidation/+15.995 Da (M). The MS/MS peptide data were exported from Xcalibur software and plotted with Origin Pro 8.5.0.

## Sample preparation for LC−MS/MS analysis

All gelatins analyzed by the LC−MS/MS method, including the single-gelatin samples, home-made adulterated DHG samples and commercial DHG products, were prepared using a simplified procedure based on a previous study [10]. First, 0.1 g of gelatin powder was added to 50 ml of a 1% $NH_4HCO_3$ solution and dissolved for 30 min by ultrasound treatment. Then, 1 ml of trypsin solution (1 mg/ml in 1% $NH_4HCO_3$, prepared immediately when needed) and 2 ml of a 1% $NH_4HCO_3$ solution were added to 1 ml of the gelatin solution. The mixture was incubated at 37˚C for 12 h and the digest was filtered through a 0.22 $\mu$m microporous membrane.

## LC−MS/MS analysis

LC−MS/MS analysis was performed using a Shimadzu Nexera X2 UHPLC system (Shimadzu, Kyoto, Japan) and an AB Sciex Q-Trap 4500 mass spectrometer equipped with a Turbo V™ source in positive mode and an electrospray ionization (ESI) source. Chromatographic separation was carried out using a Kinetex C18 100 Å column (2.6×100×4.6 mm, Phenomenex, Torrance, CA, USA) at a temperature of 40˚C. The sample injection volume was 5 $\mu L$. The mobile phase consisted of Buffer A (0.1% FA in water) and Buffer B (ACN). Gradient elution was performed as follows: 0−3 min, 10−16% Buffer B; 3−7 min, 16−50% Buffer B; 7−8 min, 50−60% Buffer B; 8−11 min, 60−90% Buffer B; 11−12 min, 90−10% Buffer B; and 12−15 min, 10% Buffer B. The flow rate was set to 0.3 ml/min. The following MS parameters were used: ion spray voltage, 4500 V; curtain gas and collision gas, 30 psi and 6 psi, respectively; ion source gas 1 and 2, 45 psi and 50 psi, respectively; and vaporizer temperature, 500˚C.

The mass spectrometer was operated in multiple reaction monitoring (MRM) mode, which allowed the simultaneous detection of several marker peptides in a single run. In our study, the ions at m/z 570.4, 386.2, 641.3 and 590.8 represented the peptides LA1, MA1, NA1 and ZA1, which are unique to DHG, HHG, CHG and PHG, respectively. These ions and their fragments formed different ion transition pairs. In the LC−MS/MS MRM experiments, each marker peptide was monitored by its corresponding ion transition pair. Before sample analysis, two or more transitions were examined for each peptide using the synthesized marker peptide standards. The ion transition with the most abundant intensity was selected to monitor the

peptide. In our study, we chose the ion transition pairs 570.4→698.3, 386.2→402.2, 641.3→726.4 and 590.8→894.5 to monitor the marker peptides LA1, MA1, NA1 and ZA1, respectively.

## Evaluation of specificity of the LC–MS/MS method

To test the specificity of the LC–MS/MS method, single-gelatin samples and a mixed standard solution containing the four marker peptides were first analyzed. After data acquisition, extracted ion chromatograms (XICs) using the ion transition pairs were generated with Analyst 1.6 software (AB Sciex). The XIC data were exported and plotted using Origin Pro 8.5.0 software. The XICs from the gelatin samples were verified by comparing the observed retention times with those determined from the standard solution. Theoretically, the four marker peptides could be detected only in their corresponding gelatin sample. Samples with an identified marker peptide indicated that the corresponding animal skin material was present.

## Evaluation of linearity, repeatability and sensitivity of the LC–MS/MS method

To evaluate the applicability of the developed method, the linearity, repeatability and sensitivity were investigated.

For the linearity tests, 10 μl of each of the four synthesized peptide standard solutions at various concentrations ranging from 0.01 μg/ml to 0.1 μg/ml were injected for LC–MS/MS analysis. Standard curves were plotted using the peak area under the XIC and the amount of sample injected. The limits of detection and quantification, defined as the peak signal with signal-to-noise ratios of 3/1 and 10/1, respectively, were determined based on the data obtained from the marker peptide standard solution injections with the lowest concentration (0.01 μg/ml).

To test the repeatability, five replicate samples of each kind of gelatin were prepared and analyzed by LC–MS/MS.

Typically, gelatin adulteration occurs when less expensive HHG, CHG or PHG are mixed into more expensive DHG. Thus, the sensitivity of a method to detect adulteration is crucial to accurately trace the gelatin source. Therefore, to evaluate the sensitivity of the method, homemade adulterated samples were prepared by mixing different amounts (0.1%, 1%, 5% and 10%) of HHG, CHG and PHG into the DHG samples. The homemade adulterated DHG samples were then analyzed to evaluate the effectiveness of this method.

Moreover, there are many different kinds of commercial DHG products on the market. During DHG processing, different excipients, such as yellow wine, crystal sugar and soybean oil, may be added [8, 11], and gelatin identification may be influenced by the excipients. Therefore, 18 batches of commercial DHG products covering a broad range of prices and divisions were collected and analyzed to test the suitability of the LC–MS/MS method and investigate the product quality.

# Results

## Theoretical marker peptide prediction

To predict the theoretical marker peptides of the four animal species, we carried out a sequence alignment of their type I collagens. As shown in Fig 1A, peptide $^{497}$GPTGEPGKPGDK$^{508}$, which is unique to donkeys, was selected because α2 (I) of the four animals has amino acid variations at positions 499, 501, 505 and 507. Likewise, as shown in Fig 1B, the amino acids at positions 423 and 424 were different among the four species. After digestion with trypsin, peptides $^{422}$GASGPAGVR$^{430}$ and $^{422}$GPTGPAGVR$^{430}$, which are unique to horses and pigs, respectively,

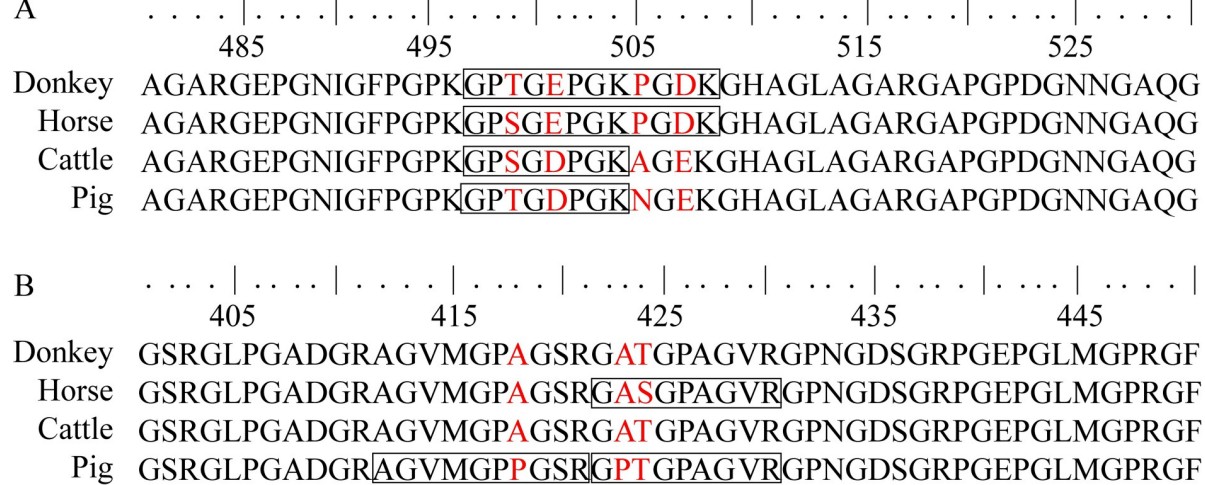

**Fig 1. Partial sequence alignments of the type I collagen α2 chains from four different animal species.** (A) Peptide unique to donkeys. (B) Peptide unique to horses. The amino acid variations are highlighted in red. Theoretical marker peptides after digestion with trypsin are indicated with black boxes.

were obtained. Another marker peptide from pig α2 (I), [412]AGVMGPPGSR[421], was obtained because of the amino acid variation at position 418. In total, 52 theoretical marker peptides were found, 18 of which were derived from the α1 chain of type I collagen and 34 of which belonged to the α2 chain of type I collagen (S2 Table). Donkeys and horses had two peptides each, while cattle and pigs had 25 and 23 marker peptides, respectively. The results provided a rationale for follow-up studies; therefore, we used LC–MS/MS to verify whether the theoretical marker peptides were present in the different gelatins.

## Marker peptides detected in gelatins

To determine if the theoretical marker peptides were present in the four kinds of gelatins, we conducted LC–MS/MS experiments. Both the α1 and α2 chains of type I collagen were determined with high confidence from database searches to possibly contain marker peptide sequences (S3 Table). Peptides from various other kinds of proteins were also identified (S4 Table). As shown in Table 1, according to the predicted marker peptides and the identification results, we found a total of 12 marker peptides (one for DHG, one for HHG, four for CHG and six for PHG) derived from type I collagen. All marker peptides were detected only in their corresponding gelatins and not in any other gelatin. Four marker peptides (ZA1~ZA4) were detected in the PHG for the first time. Peptides LA1 (Fig 2A) and MA1 (Fig 2B) were the only marker peptides for DHG and HHG, respectively. NA1 (Fig 2C) and ZA1 (Fig 2D) showed the highest abundance in CHG and PHG, respectively. Therefore, these four peptides were selected and synthesized as standards to further develop an LC–MS/MS MRM method.

## Specificity of the LC–MS/MS method

To investigate the specificity of the method, four kinds of gelatins were first tested. The results showed that the four marker peptides eluted at 1.14 min, 1.24 min, 1.44 min and 3.19 min (Fig 3). Each marker peptide was detected in only the corresponding gelatin. Therefore, the identification of each kind of gelatin could be undoubtedly confirmed by the presence of the corresponding marker peptide and the absence of the other marker peptides. These results demonstrated that this method had good specificity.

**Table 1. Marker peptides detected in different gelatins.**

| Number | Marker peptides | Collagen | Species | Length | Mass (Da) | Charge | Gelatin |
|--------|-----------------|----------|---------|--------|-----------|--------|---------|
| LA1 | [497]GPTGEPGKPGDK[508] | α2(I) | Donkeys | 12 | 1139.5800 | 2 | DHG |
| MA1 | [422]GASGPAGVR[430] | α2(I) | Horses | 9 | 770.4035 | 2 | HHG |
| NA1 | [781]GEAGPSGPAGPTGAR[795] | α1(I) | Cattle | 15 | 1280.6109 | 2 | CHG |
| NA2 | [1066]GETGPAGPAGPIGPVGAR[1083] | α1(I) | Cattle | 18 | 1559.8056 | 2; 3 | CHG |
| NA3 | [829]SGETGASGPPGFVGEK[844] | α2(I) | Cattle | 16 | 1475.6892 | 2 | CHG |
| NA4 | [1066]IGQPGAVGPAGIR[1078] | α2(I) | Cattle | 13 | 1191.6724 | 2 | CHG |
| ZA1 | [976]TGQPGAVGPAGIR[988] | α2(I) | Pigs | 13 | 1179.6360 | 2; 3 | PHG |
| ZA2 | [313]GRPGPPGPAGAR[324] | α1(I) | Pigs | 12 | 1088.5839 | 2; 3 | PHG |
| ZA3 | [624]QGPSGPSGER[633] | α1(I) | Pigs | 10 | 970.4468 | 2 | PHG |
| ZA4 | [422]GPTGPAGVR[430] | α2(I) | Pigs | 9 | 810.4348 | 2 | PHG |
| ZA5 | [1069]GETGPAGPAGPVGPVGAR[1086] | α1(I) | Pigs | 18 | 1549.7900 | 2; 3 | PHG |
| ZA6 | [739]TGETGASGPPGFAGEK[754] | α2(I) | Pigs | 16 | 1461.6736 | 2 | PHG |

The sequence number corresponds to the actual position of the collagen sequence from the UniProt database.

α1(I) and α2(I) indicate the α1 and α2 polypeptide chains of type I collagen.

## Linearity of the synthetic marker peptides

Mixed standards of the four synthesized marker peptides at various concentrations were analyzed to test the linearity of the LC–MS/MS MRM method. The results showed that all $R^2$ values were greater than 0.99 (S1 Fig), indicating a good correlation between the peak area and amount of each marker peptide injected. In addition, the limit of detection–limit of quantification of LA1, MA1, NA1 and ZA1 were 7.14 pg– 23.56 pg, 13.62 pg– 44.95 pg, 2.46 pg– 8.12 pg and 1.85 pg– 6.11 pg, respectively.

## Repeatability of the identification of the four gelatins

To confirm whether the LC–MS/MS MRM method provided reproducible detection of the marker peptides, five independent replicate experiments for each gelatin were performed. The five replicates showed small differences in retention time and peak intensity (S2 Fig), indicating that the repeatability of this method was satisfactory.

## Sensitivity of the LC–MS/MS method

Homemade mixed gelatin samples were analyzed to test the ability of the method to identify adulterated DHGs. As shown in S3 Fig, when 0.1% of the three other gelatins were added, the DHG, CHG and PHG marker peptides were all easily observed. This result indicated that the addition of CHG and PHG could be detected at a level of 0.1%. HHG was accurately identified at a level of 0.5%. These results suggested that the LC–MS/MS MRM method developed in our study is sufficiently sensitive to detect animal skin materials in adulterated DHGs.

## Identification of commercial DHG products

A broad range of commercial DHG products were tested to confirm the feasibility of the method for practical applications. From a total of 18 batches of products, nine were identified as authentic DHG, as only LA1 was detected (S4A Fig). However, MA1 was found in eight batches of the samples (S4B and S4D Fig). Thus, there is a strong possibility that these samples were adulterated with horse skin materials. In addition, NA1 was detected in two of the

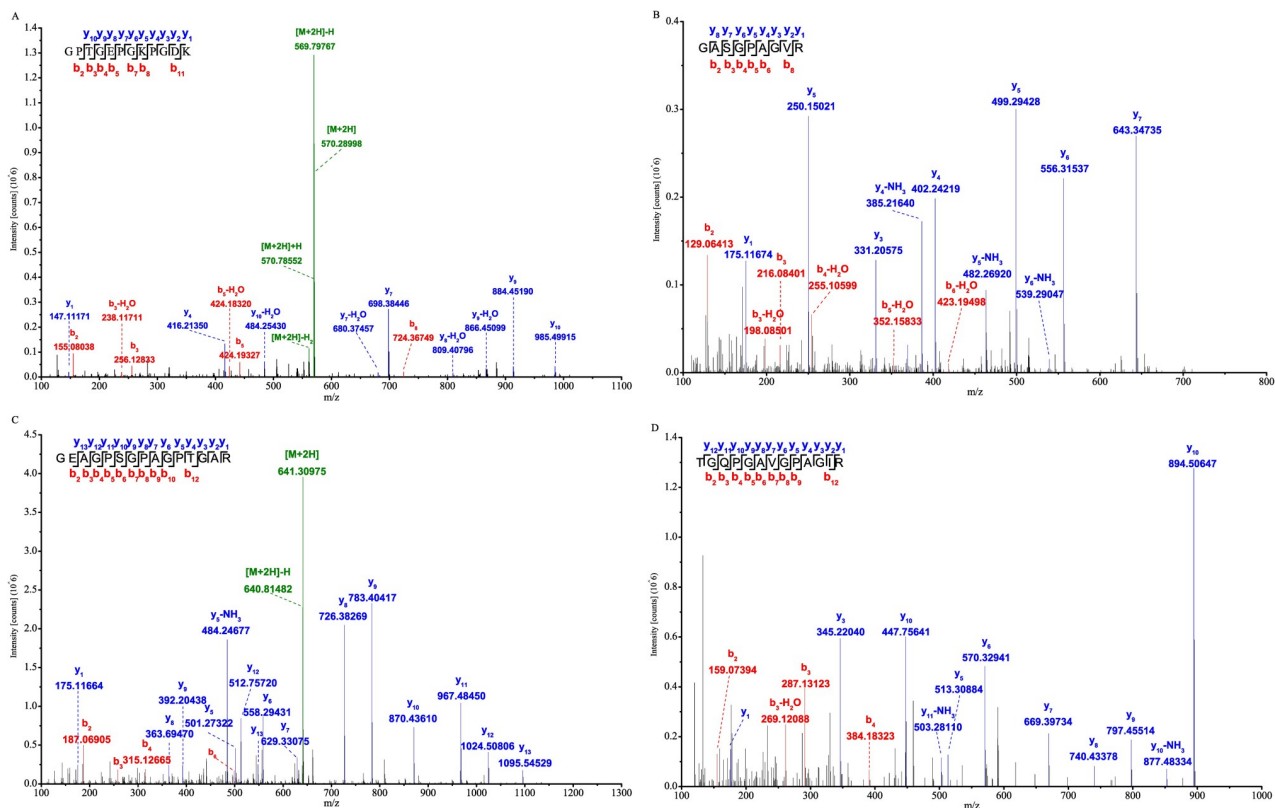

**Fig 2. MS/MS spectra of marker peptides detected in the four digested gelatins.** Peptides (**A**) [497]GPTGEPGKPGDK[508], (**B**) [422]GASGPAGVR[430], (**C**) [781]GEAGPSGPAGPTGAR[795] and (**D**) [976]TGQPGAVGPAGIR[988] were detected in donkey-hide, horse-hide, cattle-hide and pig-hide gelatins, respectively. The b and y ions shown in red and blue represent the prefix (N-terminal b-ion) and the suffix (C-terminal y-ion) fragments of the given peptide, respectively. X axis and Y axis represent the m/z and the intensity of the ion fragments, respectively.

products (S4C and S4D Fig), which were suspected to be adulterated with cattle skin materials. Interestingly, ZA1 was not found in any commercial DHG products collected in our study.

## Discussion

DHG is a widely used traditional Chinese medicine, but its identification has puzzled researchers for decades. In this study, we investigated whether there are marker peptides in different gelatins and whether these peptides could be used to identify gelatins. To identify the marker peptides, we performed bioinformatics analysis for their prediction followed by proteomic experiments for verification. We found 12 marker peptides in the four gelatins. Based on the detected marker peptides, we developed an LC–MS/MS method and then examined its applicability for both single and mixed gelatin samples. Our results demonstrated that the developed method had good specificity, repeatability and sensitivity.

By sequence alignment analysis, we predicted 52 theoretical marker peptides derived from the type I collagens of the four animals. A previous study [8] showed certain different theoretical marker peptides from ours. However, in contrast to this previous study, we used both the α1 (I) and α2 (I) chains of the four animals for the prediction, which may account for the different results. Our results also showed that most of the peptides were from CHG and PHG. DHG and HHG each had only two predicted peptides because donkeys are closely related to horses, so there are few amino acid variations among their collagen sequences.

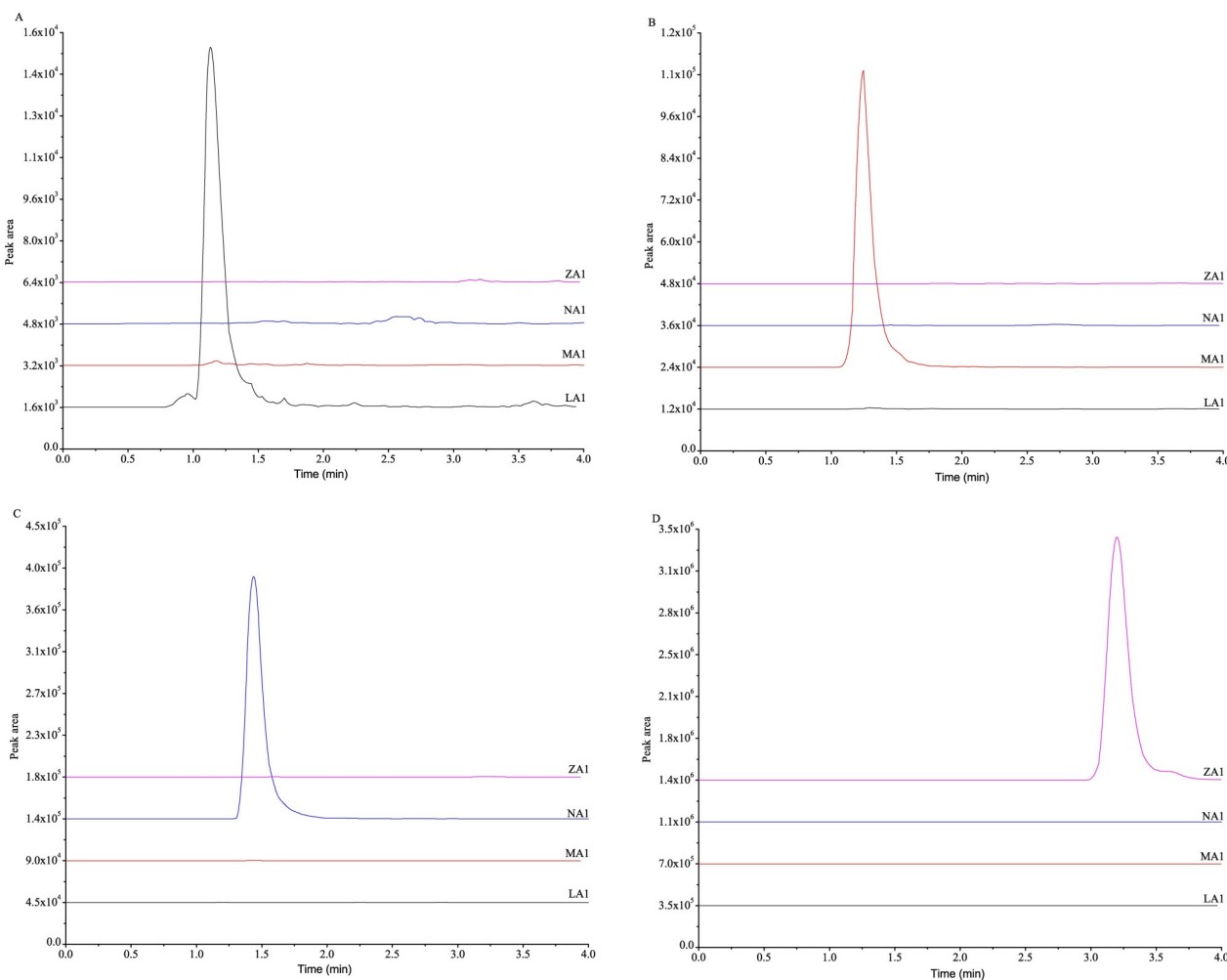

**Fig 3. Detection of marker peptides in the four kinds of gelatin by LC–MS/MS MRM. (A)** Donkey-hide gelatin, **(B)** horse-hide gelatin, **(C)** cattle-hide gelatin, and **(D)** pig-hide gelatin. LA1, MA1, NA1 and ZA1 are the four marker peptides. All marker peptides were detected only in the corresponding gelatin.

After proteomic analysis verification, we found 12 marker peptides in the four kinds of gelatin. Some of the peptides were consistent with those from other relevant studies [8, 10, 17]. These findings should help us distinguish among different gelatins. In addition, four marker peptides from PHG that were detected for the first time in our study may be used as possible substitutes to identify pig materials in the future. However, 40 predicted peptides, which accounted for 77% of the total peptides, were not detected in our experiments. Previous studies [1] conducted by other researchers on the components in gelatins may explain this result. Proteins and peptides may suffer from denaturation due to the gelatin heating process. Thus, the peptides that were not detected may have been destroyed during gelatin formation.

To develop an LC–MS/MS method with good specificity, selecting marker peptides is important. The highly similar peptides produced by each of the four gelatins may disrupt the identification of the other peptides [10]. In the Chinese Pharmacopoeia 2015 edition, the peptide GPPGAAGPPGPR was used to identify DHG [19]. However, according to a previous study, the peptide GPPGAAGPPGPR is derived from type I collagen and is common to both donkeys and horses, and it can therefore be detected in both DHG and HHG [9]. Thus, in our

study, we selected different marker peptides for each gelatin to establish an LC–MS/MS method. Next, we tested the specificity of the method. Our results showed that the four selected marker peptides could be clearly identified without any interference. Moreover, we examined the linearity, repeatability and sensitivity of the method, and the results demonstrated that it has good applicability for the identification of gelatins.

We collected 18 batches of commercial DHG products and used the LC–MS/MS method to identify them. Surprisingly, almost half of these samples were suspected to be adulterated with horse skin materials, revealing the current state of DHG quality control. Donkeys are more closely related to horses than to cattle or pigs. The collagen sequences between donkeys and horses are therefore very similar, making it extremely difficult to distinguish between them [1, 22]. This may explain why DHGs were particularly prone to adulteration with HHG. Notably, the LC–MS/MS method developed in our study may be useful for addressing this issue.

In conclusion, we successfully predicted and verified marker peptides in gelatins. In addition, we developed an LC–MS/MS method with good specificity, repeatability and sensitivity. These findings might help us to identify these four kinds of gelatin and their adulterants. Despite our promising results, questions remain. In recent years, DHG products made from donkey meat or bones have begun to appear on the market. These DHG products are of poor quality because of the insufficient amount of donkey skin. In the Chinese Pharmacopoeia 2020 edition, poor-quality DHG products are identified by quantitative determination of the marker peptides [23], and an insufficient amount of donkey skin may lead to a low level of marker peptides. However, marker peptides can be synthesized and added to DHG products to satisfy the detection criterion. Therefore, DHG products with an insufficient amount of donkey skin cannot be completely avoided by quantitatively measuring the amounts of marker peptides. Further studies that address these problems will be needed.

## Conclusions

The purpose of this study was to identify four gelatins by LC–MS/MS using marker peptides. Herein, we successfully predicted theoretical marker peptides of four gelatins, and a total of 12 marker peptides for the four gelatins were verified by proteomic experiments. Furthermore, we developed an LC–MS/MS method for peptide marker detection, which was applicable and demonstrated good specificity and sensitivity for the identification of gelatins. These results can help us control the gelatin quality. In addition, the strategy of predicting theoretical marker peptides followed by their verification via proteomic experiments can be adapted to other medicines containing protein components, especially known proteins.

## Supporting information

**S1 Fig.** Linearity of the LC–MS/MS MRM method for the analysis of the four marker peptides: (A) 570.4→698.3 for LA1, (B) 386.2→402.2 for MA1, (C) 641.3→726.4 for NA1, and (D) 590.8→894.5 for ZA1.
(TIF)

**S2 Fig. Repeatability evaluation of the established LC–MS/MS MRM method. (A)** Donkey-hide gelatin, **(B)** horse-hide gelatin, **(C)** cattle-hide gelatin, and **(D)** pig-hide gelatin. Five replicates for each gelatin sample were measured independently, and the results were reproducible.
(TIF)

**S3 Fig. Identification of donkey-hide gelatins adulterated with different amounts of the other three gelatins.** Donkey-hide gelatin samples were mixed with the following amounts of horse-hide gelatin, cattle-hide gelatin and pig-hide gelatin: **(A)** 0.1%, **(B)** 0.5%, **(C)** 5.0%, and

**(D)** 10.0%. The addition of two of the three other gelatins could be detected at a level of 0.1% of the total weight.
(TIF)

**S4 Fig. Detection of commercial donkey-hide gelatin products. (A)** Identified as an authentic donkey-hide gelatin product. Samples **(B)**, **(C)** and **(D)** were identified as counterfeit commercial donkey-hide gelatin products adulterated with horse-hide gelatin and cattle-hide gelatin.
(TIF)

**S1 Table. Information on the samples used in our study.**
(XLSX)

**S2 Table. All predicted theoretical marker peptides of the four species.**
(XLSX)

**S3 Table. Identification of proteins from the four kinds of gelatins.**
(XLSX)

**S4 Table. Identification of peptides from the four kinds of gelatins.**
(XLSX)

## Author Contributions

**Conceptualization:** Menghua Wu, Zhiguo Ma, Ying Zhang, Hui Cao.

**Funding acquisition:** Hui Cao.

**Investigation:** Jinju Zhang, Ying Zhang.

**Methodology:** Jinju Zhang.

**Project administration:** Hui Cao.

**Software:** Jinju Zhang.

**Supervision:** Menghua Wu, Zhiguo Ma, Ying Zhang, Hui Cao.

**Visualization:** Zhiguo Ma.

**Writing – original draft:** Jinju Zhang.

**Writing – review & editing:** Menghua Wu, Zhiguo Ma, Ying Zhang, Hui Cao.

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
