## [Decision Letter · Decision Letter 0]

28 Jun 2022

PONE-D-22-11303Species-specific identification of donkey-hide gelatin and its adulterants using marker peptidesPLOS ONE

Dear Dr. Zhang,

Thank you for submitting your manuscript to PLOS ONE. After careful consideration, we feel that it has merit but does not fully meet PLOS ONE’s publication criteria as it currently stands. Therefore, we invite you to submit a revised version of the manuscript that addresses the points raised during the review process.

Comments: Please carefully revise the language in the text, paying particular attention to the format of the references.

We look forward to receiving your revised manuscript.

Kind regards,

Chun-Hua Wang

Academic Editor

PLOS ONE

Journal Requirements:

"This work was supported by Guangdong Natural Science Foundation (No. 2020A151501577, Ying Zhang) and the Open Fund of Guangdong Province Key Laboratory of Pharmacodynamic Constituents of TCM and New Drugs Research (No. 2016B 030301004, Hui Cao). YZ and HC conceived and designed the study."

Reviewers' comments:

Reviewer's Responses to Questions

**Comments to the Author**

1. Is the manuscript technically sound, and do the data support the conclusions?

Reviewer #1: Yes

Reviewer #2: Yes

Reviewer #3: Yes

Reviewer #4: Partly

2. Has the statistical analysis been performed appropriately and rigorously? 

Reviewer #1: Yes

Reviewer #2: Yes

Reviewer #3: I Don't Know

Reviewer #4: Yes

3. Have the authors made all data underlying the findings in their manuscript fully available?

Reviewer #1: Yes

Reviewer #2: Yes

Reviewer #3: Yes

Reviewer #4: No

4. Is the manuscript presented in an intelligible fashion and written in standard English?

Reviewer #1: Yes

Reviewer #2: Yes

Reviewer #3: Yes

Reviewer #4: No

5. Review Comments to the Author

Reviewer #1: 1.Only Standard medicine samples DHG, CHG and PHG were purchased from from the National Institutes for Food and Drug Control (NIFDC; Beijing, China), but HHG was not purchased. Please explain the reasons.

2.Can the doping amount in donkey-hide gelatin be identified according to the existing marker peptides content determination results?

3.What is the purity of the synthetic marker peptide standard?

4.Is there really enough horse hide for adulteration in donkey-hide gelatin production?Is it of practical significance to identify the adulterated horse hide in donkey-hide gelatin?

Reviewer #2: Donkey-hide gelatin is an important traditional Chinese medicine made from donkey skin. Identifying the animal materials (donkeys, horses, cattle and pigs) in donkey-hide gelatin is challenging. The goal of this study by Zhang J. et al. is to utilize liquid chromatography–tandem mass spectrometry method to identify species-specific marker peptides, and these marker peptides can be used to discriminate the ingredient from (donkeys, horses, cattle and pigs). This reviewer realizes that the results were so interesting because the marker peptides and LC–MS/MS method showed a good specificity, sensitivity and reproducibility. This method is valuable in controlling the quality of donkey-hide gelatin. My comments to co-authors are:

1. Figure 1 and the words that describe Figure 1 from theoretical marker peptide prediction section in Materials and Methods should be in Result section, and the Figure 1a should be Figure 1A.

2. In the “Evaluation of applicability of the LC–MS/MS method” section and the corresponding result section, the authors should organize the word and data to highlight the specificity, repeatability and sensitivity of the method.

3. In the legend of Figure 2, the authors should describe the Figure in details, such as, what are the meanings of b and y? different colors? X axis and Y axis.

4. In line 303, R2 should be R2.

5. In line 366 and 367, “Unlike our study, a previous study showed certain different theoretical marker peptides.” The authors should provide references for the study.

6. In line 379 and 380, “Previous studies conducted by other researchers on the components in gelatins may explain this result.” The authors should provide references for these studies.

7. In the manuscript, “in this study” should be changed as “in our study”.

8. In the manuscript, S1 Figure should be Figure S1 and S1 Table should be Table S1.

Reviewer #3: This is a well-written paper containing interesting results which merit publication. The paper presents an LC-MS/MS method for peptide marker detection, which was applicable and demonstrated good specificity and sensitivity for the identification of gelatins. It is a topic of interest to the researchers in the related areas but the manuscript needs a bit improvement (minor revision) before acceptance for publication. My detailed comments are as follows:

1. Please note that the English letters marked on the chart in the article are consistent with those marked on the chart itself, which can be in lowercase or uppercase.

2. Fig4 can also be attached to the end of the article like S3 figure.

3. The consistency between the information displayed by S3 figure and the content described in the discussion part of the article needs to be improved. （MAI exists in samples（b）、（d），instead（b）、（c）.

4. If the marker peptide was added into HHG, the proposed LC-MS/MS method can not identify the HHG. Please try to discuss this point deeply and try to draw a possible method to assistant to identify the it.

Reviewer #4: 1． There have been some research papers on identification of donkey-hide glue and its adulterants, so this paper should show the novelty of the work in the Introduction section, compared with the literature that had been reported. I do not think it is correct that the authors stated this manuscript as “one of the first investigations to focus specifically on the identification of gelatins using marker peptides”. Please add the literatures of related studies in the introduction part.

2． Dithiothreitol often was used to destroy the disulfide bond, does the collagen contain disulfide bond? Please make sure the necessity of Dithiothreitol must be used to destroy the disulfide bond?

3． From the manuscripts, 12 marker peptides for the four gelatins were verified only by proteomic experiments, it advised that the 12 marker peptides should be validated by different sources samples.

4． It is recommended that the manuscripts be professionally edited for English language before submitting your revised manuscript version.

6. PLOS authors have the option to publish the peer review history of their article (what does this mean?). If published, this will include your full peer review and any attached files.

Reviewer #1: **Yes: **Zou Xiaoxing

Reviewer #2: No

Reviewer #3: **Yes: **Yuming Dong

Reviewer #4: **Yes: **WEI Feng

---

## [Author Response · Author response to Decision Letter 0]

11 Jul 2022

Responses to comments about Journal Requirements

Comment #1: Please ensure that your manuscript meets PLOS ONE's style requirements, including those for file naming.

Response #1: We have carefully checked the paper to ensure that the manuscript meets PLOS ONE's style requirements.

Comment #2: Thank you for stating the following financial disclosure…Please state what role the funders took in the study. If the funders had no role, please state: "The funders had no role in study design, data collection and analysis, decision to publish, or preparation of the manuscript."

Response #2: This work was supported by Guangdong Natural Science Foundation (No. 2020A151501577, Ying Zhang), the Open Fund of Guangdong Province Key Laboratory of Pharmacodynamic Constituents of TCM and New Drugs Research (No. 2016B030301004, Hui Cao), and Cao Hui’s Expert Workstation in Yunnan Province (202005AF150011, Hui Cao). The funder of Ying Zhang and Hui Cao had contributed to the project administration, supervision, validation, draft writing and editing throughout the period of this research. In order to make it more clearly about all author's contributions, we add the part of “Author Contributions” in the manuscript (Page 22, Line 464-475).

Comment #3: We note that you have included the phrase “data not shown” in your manuscript. Unfortunately, this does not meet our data sharing…we ask that you remove the phrase that refers to these data.

Response #3: We are sorry for neglecting the requirements of data sharing. For the first section of “Data not shown”, we have removed the related data (Page 16, Line 342-344), because it is not a core part of the research. For the second section of “Data not shown”, we have added a reference to support our points (Page 20, Line 417-418).

Comment #4: Please review your reference list to ensure that it is complete and correct….

Response #4: We have reviewed the references list and made some changes according to the comments of reviewers (Page 23-27, Line 477-571).

Responds to comments from the reviewers

Reviewer #1:

Comment #1: Only Standard medicine samples DHG, CHG and PHG were purchased from the National Institutes for Food and Drug Control (NIFDC; Beijing, China), but HHG was not purchased. Please explain the reasons.

Response #1: The HHG made by horse skins cannot be used as food stuffs or medicines, although horse skins were always adulterated into donkey skins to make DHG. There is no reference substance of HHG provided by NIFDC or sold on the market. So, we used the house-made HHG samples to perform the proteomics study and build the LC-MS/MS method.

Comment #2: Can the doping amount in donkey-hide gelatin be identified according to the existing marker peptides content determination results?

Response #2: This is a really interesting question and we have also taken this possibility into account. But unfortunately, the doping amount in donkey-hide gelatin cannot be identified based on the results. We will focus on this question and try to solve this problem.

Comment #3: What is the purity of the synthetic marker peptide standard?

Response #3: The purity of the synthetic marker peptide standard is more than 95%. We have added it in the manuscript (Page 5, Line 106).

Comment #4: Is there really enough horse hide for adulteration in donkey-hide gelatin production? Is it of practical significance to identify the adulterated horse hide in donkey-hide gelatin?

Response #4: Since the donkey-hide gelatin was recorded in Chinese Pharmacopoeia (1977 edition), the great efforts have been made to push the identification of donkey-hide gelatin and other gelatins, including horse-hide gelatin. Based on the identification results of the commercial DHG products, almost half of all samples were adulterated with horse skin materials, which indicated that the identification of the adulterated horse hide in donkey-hide gelatin is of great practical significance.

Reviewer #2:

Comment #1: Figure 1 and the words that describe Figure 1 from theoretical marker peptide prediction section in Materials and Methods should be in Result section, and the Figure 1a should be Figure 1A.

Response #1: Thanks for your good comments. We have moved the Figure 1 and the words that describe Figure 1to the Results section (Page 13, Line 266-274). We have changed the Figure 1a to Figure 1A (Page 13, Line 266-284) and corrected other same mistakes. 

Comment #2: In the “Evaluation of applicability of the LC–MS/MS method” section and the corresponding result section, the authors should organize the word and data to highlight the specificity, repeatability and sensitivity of the method.

Response #2: According to the comments of reviewers, we have reorganized the words and data to make it more readable (Page 11-13, Line 221-251). And we also changed the subtitle of the “Evaluation of applicability of the LC–MS/MS method” section and the corresponding results section to highlight the specificity, repeatability and sensitivity of the method (Page 11, Line 221-222; Page 11, Line 232-233; Page 15, Line 313-314; Page 17, Line 351-352). 

Comment #3: In the legend of Figure 2, the authors should describe the Figure in details, such as, what are the meanings of b and y? different colors? X axis and Y axis. 

Response #3: Thanks for your good comments. The MS/MS spectrum contains a complex mixture of peptide fragments. The b and y ions shown in red and blue represent the prefix (N-terminal b-ion) and the suffix (C-terminal y-ion) fragments of the given peptide, respectively. The X axis and Y axis represent the m/z and the intensity of the ion fragments, respectively. We have added this description to the legend of Figure 2 (Page 15, Line 309-312).

Comment #4: In line 303, R2 should be R2.

Response #4: We are sorry about it and have corrected this mistake (Page 16, Line 329).

Comment #5: In line 366 and 367, “Unlike our study, a previous study showed certain different theoretical marker peptides.” The authors should provide references for the study.

Response #5: We have added the corresponding references for the study (Page 19, Line 393).

Comment #6: In line 379 and 380, “Previous studies conducted by other researchers on the components in gelatins may explain this result.” The authors should provide references for these studies.

Response #6: We have added the corresponding references for the study (Page 19, Line 406).

Comment #7: In the manuscript, “in this study” should be changed as “in our study”.

Response #7: We have changed “in this study” to “in our study” in the manuscript.

Comment #8: In the manuscript, S1 Figure should be Figure S1 and S1 Table should be Table S1.

Response #8: We have changed the file naming according the PLOS ONE style templates to ensure that our manuscript meets PLOS ONE's style requirements.

Reviewer #3:

Comment #1: Please note that the English letters marked on the chart in the article are consistent with those marked on the chart itself, which can be in lowercase or uppercase.

Response #1: We are sorry about this and we have modified the letters to ensure that the letters marked on the chart in the article are consistent with those marked on the chart itself.

Comment #2: Fig4 can also be attached to the end of the article like S3 figure.

Response #2: Thanks for your good comments. We have changed the Figure 4 to Figure S3 and renumbered other figures (Page 17, Line 361). It makes the manuscript become more readable.

Comment #3: The consistency between the information displayed by S3 figure and the content described in the discussion part of the article needs to be improved. (MAI exists in samples (b), (d), instead (b), (c).

Response #3: We are so sorry about this mistake. And we have corrected this mistake (Page 18, Line 369-373).

Comment #4: If the marker peptide was added into HHG, the proposed LC-MS/MS method cannot identify the HHG. Please try to discuss this point deeply and try to draw a possible method to assistant to identify it.

Response #4: This is a really interesting question. Thank you so much for your good comments. In our study, each marker peptide for one gelatin was selected to build the LC-MS/MS method. Theoretically, the four marker peptides could only be detected in their corresponding gelatin samples. Samples with an identified marker peptide was well related to the corresponding animal skin material. That is to say, only marker peptide MA1 should be detected in HHG samples, and samples detected with multiple marker peptides were considered as adulterated samples. If marker peptides of LA1, NA1 and ZA1 are added into HHG, these marker peptides including MA1 will be detected and the samples will be considered as adulterated samples. Based on the identification results, we could not determine whether the marker peptides of LA1, NA1 and ZA1 were derived from the corresponding samples or just artificially added. To solve this problem, we have considered whether proteins which are difficult to synthesize could be used to identify different gelatins. The Parallel Reaction Monitoring (PRM) which is commonly used to quantify proteins may be used to analysis the collagens or other proteins in different gelatins. If we establish a PRM method which can specifically identify the HHG based on the protein components, the problem may be solved brilliantly.

Reviewer #4:

Comment #1: There have been some research papers on identification of donkey-hide glue and its adulterants, so this paper should show the novelty of the work in the Introduction section, compared with the literature that had been reported. I do not think it is correct that the authors stated this manuscript as “one of the first investigations to focus specifically on the identification of gelatins using marker peptides”. Please add the literatures of related studies in the introduction part.

Response #1: Thanks for your good comments. Previous studies have focus on the identification of donkey-hide gelatin using the marker peptides method. We have deleted the statement of “one of the first investigation…” and added the related studies in the introduction section (Page 4, Line 68 and 87). And in our study, we used both the α1 (I) and α2 (I) chains to predict the marker peptides of four animals. Then we verified the marker peptides and applied them to build an MRM method. Based on the methods, we also collected and tested kinds of commercial donkey-hide gelatin products. It is a comprehensive study of the identification of donkey-hide gelatin, and we believed that our study can help improve the quality control of donkey-hide gelatin.

Comment #2: Dithiothreitol often was used to destroy the disulfide bond, does the collagen contain disulfide bond? Please make sure the necessity of Dithiothreitol must be used to destroy the disulfide bond?

Response #2: Type I collagens contain two α1 chains [α1 (I)] and one α2 chain [α2 (I)], and there are disulfide bonds between chains of collagens. According to previous studies, dithiothreitol was always used during the sample’s preparation for the proteomics analysis. In our study, not only the collagens but also other kinds of proteins from four gelatins were used to the proteomics study. So, it is necessary to use dithiothreitol for samples’ preparation in our study.

Comment #3: From the manuscripts, 12 marker peptides for the four gelatins were verified only by proteomic experiments, it advised that the 12 marker peptides should be validated by different sources samples.

Response #3: Thanks for your good comments. Donkey-hide gelatin are most commonly adulterated with materials from horse, pig and cattle. However, it is also possible that donkey-hide gelatin will be adulterated with tissues from sheep, deer or other animals. According to the Introduction section in the manuscript, we find that previous studies had tried to identify the donkey-hide gelatin and other gelatin like deer-horn glue, etc. In fact, it is a really good idea to validate the 12 marker peptides using other kinds of samples in our further study. Thanks again for your good advice.

Comment #4: It is recommended that the manuscripts be professionally edited for English language before submitting your revised manuscript version.

Response #4: We are very sorry about this. The revised manuscript version has been edited for proper English language, grammar, punctuation, spelling, and overall style by one or more of the highly qualified native English-speaking editors in American Journal Experts (AJE) corporation.

---

## [Decision Letter · Decision Letter 1]

18 Jul 2022

PONE-D-22-11303R1Species-specific identification of donkey-hide gelatin and its adulterants using marker peptidesPLOS ONE

Dear Dr. Zhang,

Thank you for submitting your manuscript to PLOS ONE. After careful consideration, we feel that it has merit but does not fully meet PLOS ONE’s publication criteria as it currently stands. Therefore, we invite you to submit a revised version of the manuscript that addresses the points raised during the review process.

We look forward to receiving your revised manuscript.

Kind regards,

Chun-Hua Wang

Academic Editor

PLOS ONE

Journal Requirements:

Reviewers' comments:

Reviewer's Responses to Questions

**Comments to the Author**

1. If the authors have adequately addressed your comments raised in a previous round of review and you feel that this manuscript is now acceptable for publication, you may indicate that here to bypass the “Comments to the Author” section, enter your conflict of interest statement in the “Confidential to Editor” section, and submit your "Accept" recommendation.

Reviewer #2: All comments have been addressed

Reviewer #3: (No Response)

2. Is the manuscript technically sound, and do the data support the conclusions?

Reviewer #2: Yes

Reviewer #3: (No Response)

3. Has the statistical analysis been performed appropriately and rigorously? 

Reviewer #2: Yes

Reviewer #3: (No Response)

4. Have the authors made all data underlying the findings in their manuscript fully available?

Reviewer #2: Yes

Reviewer #3: (No Response)

5. Is the manuscript presented in an intelligible fashion and written in standard English?

Reviewer #2: Yes

Reviewer #3: (No Response)

6. Review Comments to the Author

Reviewer #2: The authors have addressed my comments and concerns. I recommend that editors accept the submission.

Reviewer #3: The authors have answered the questions well.However, there are still some other questions to be considered.

1. If this LC-MS/MS of marker peptide is not the first investigation on DHG and its adulterants(The study presented in this thesis is a comprehensive one of the first investigations to focus specifically on the identification of gelatins using marker peptides.),what is the innovation of this LC-MS/MS?Please compare the proposed method and the published LC-MS/MS method of marker peptide on the identification of DHG.

2. Please add the method of identification of DHG of Chinsed Pharmaceopoeia 2020 version and discuss the limitations of the CHP2020 method briefly and show the proposed LC-MS/MS can be helpful to improve the CHP 2020 method in Introducation part but not in the discussion part.

3. Line113-115, … Chinese Pharmacopoeia 114 and the quality of every gelatin sample complied with Chinese Pharmacopeia 115 requirements .Please add the version of the Chinese Pharmacopoeia .for example,2020 version.

7. PLOS authors have the option to publish the peer review history of their article (what does this mean?). If published, this will include your full peer review and any attached files.

Reviewer #2: No

Reviewer #3: **Yes: **OK

---

## [Author Response · Author response to Decision Letter 1]

20 Jul 2022

Responses to comments about Journal Requirements

Comment #1: Please review your reference list to ensure that it is complete and correct…and full reference for the retraction notice.

Response #1: We have carefully reviewed the references list to ensure that the references meet PLOS ONE's requirements.

Responds to comments from the reviewers

Reviewer #3:

Comment #1: If this LC-MS/MS of marker peptide is not the first investigation on DHG and its adulterants (The study presented in this thesis is a comprehensive one of the first investigations to focus specifically on the identification of gelatins using marker peptides.), what is the innovation of this LC-MS/MS? Please compare the proposed method and the published LC-MS/MS method of marker peptide on the identification of DHG.

Response #1: Thanks for your good comments. As you mentioned before, previous studies have focus on the identification of donkey-hide gelatin based on the marker peptides. So, we have deleted the statement of “one of the first investigation…”. LC-MS/MS based on proteomics experiments are commonly used on the discovery of marker peptides, meanwhile the reference protein database used to predict the theoretical marker peptides and specified search is essential for identifying peptides. After the discovery of marker peptides, it is also a challenge to build a fast, sensitive and accurate method based on the marker peptides. In our study, to obtain enough candidate marker peptides, we used both the α1 (I) and α2 (I) chains to predict the theoretical marker peptides. Although we focused on the marker peptides derived from type I collagen, we used all the reference proteins of four animals from UniProt as searched database. Our results might provide a reference to discover other marker peptides which derived from non-collagen proteins such as keratins. Using the marker peptides, we build an MRM method which allows a fast identification of four kinds of gelatins in 15 minutes with good specialty and sensitivity. We also collected and tested a variety kinds of commercial donkey-hide gelatin products which covers large brands, and the results might help us understand the real situation of the donkey-hide gelatin on the market.

Comment #2: Please add the method of identification of DHG of Chinsed Pharmacopoeia 2020 version and discuss the limitations of the CHP2020 method briefly and show the proposed LC-MS/MS can be helpful to improve the CHP 2020 method in Introducation part but not in the discussion part.

Response #2: This is an interesting question. In Chinese Pharmacopoeia 2020 edition, quantitative determination of A1 and A2 peptides derived from donkey skins are used to assess the quality of donkey-hide gelatin. The total amount of A1 and A2 should reach at least 0.15%. The donkey-hide gelatin with a low level of A1 and A2 may be considered as products with an insufficient amount of donkey skin. However, marker peptides of A1 and A2 can be synthesized and added to donkey-hide gelatin products to satisfy the detection criterion. Therefore, donkey-hide gelatin products with an insufficient amount of donkey skin cannot be completely avoided only by quantitative analysis of the amounts of marker peptides. Unfortunately, not only our current research but also other studies cannot provide a good solution to avoid the artificial addition of the marker peptides. We've been paying attention and trying to solve this problem in the future. Alternatively, it may be more appropriate to discuss this issue in the “Discussion” section. We have considered whether proteins which are difficult to synthesize could be used to identify different gelatins. The Parallel Reaction Monitoring (PRM) which is commonly used to quantify proteins may be used to analysis the collagens or other proteins in different gelatins. If we establish a PRM method which can specifically identify the donkey-hide gelatin based on the quantitative determination of protein components, the problem may be solved satisfactorily at last.

Comment #3: Line113-115, … Chinese Pharmacopoeia 114 and the quality of every gelatin sample complied with Chinese Pharmacopeia 115 requirements. Please add the version of the Chinese Pharmacopoeia. for example,2020 version.

Response #3: We have added the version of Chinese Pharmacopoeia in the manuscript (Page 6, Line 112-113).

---

## [Editor Report · Decision Letter 2]

2 Aug 2022

Species-specific identification of donkey-hide gelatin and its adulterants using marker peptides

PONE-D-22-11303R2

Dear Dr. Zhang,

We’re pleased to inform you that your manuscript has been judged scientifically suitable for publication and will be formally accepted for publication once it meets all outstanding technical requirements.

Kind regards,

Chun-Hua Wang

Academic Editor

PLOS ONE

---

## [Editor Report · Acceptance letter]

4 Aug 2022

PONE-D-22-11303R2 

Species-specific identification of donkey-hide gelatin and its adulterants using marker peptides 

Dear Dr. Zhang:

I'm pleased to inform you that your manuscript has been deemed suitable for publication in PLOS ONE. Congratulations! Your manuscript is now with our production department. 

Kind regards, 

on behalf of

Dr. Chun-Hua Wang 

Academic Editor

PLOS ONE